# Unbiased Watermark for Large Language Models

## Abstract

The recent advancements in large language models (LLMs) have sparked a growing
apprehension regarding the potential misuse. One approach to mitigating this risk
is to incorporate watermarking techniques into LLMs, allowing for the tracking and
attribution of model outputs. This study examines a crucial aspect of watermark-
ing: how significantly watermarks impact the quality of model-generated outputs.
Previous studies have suggested a trade-off between watermark strength and out-
put quality. However, our research demonstrates that it is possible to integrate
watermarks without affecting the output probability distribution with appropriate
implementation. We refer to this type of watermark as an **unbiased watermark**.
This has significant implications for the use of LLMs, as it becomes impossible
for users to discern whether a service provider has incorporated watermarks or not.
Furthermore, the presence of watermarks does not compromise the performance
of the model in downstream tasks, ensuring that the overall utility of the language
model is preserved. Our findings contribute to the ongoing discussion around
responsible AI development, suggesting that unbiased watermarks can serve as
an effective means of tracking and attributing model outputs without sacrificing
output quality.

## 1   Introduction

In recent years, large language models (LLMs) [19, 39, 40] have become an indispensable tool for a
wide range of tasks, including text generation [27, 10], translation [7, 5], summarization [36], etc.
With the escalating misuse of LLMs, such as plagiarism, tracking the usage of text generated by
machines has become increasingly important. One viable method to monitor the usage of LLMs
is watermarking [20, 32, 59], which embeds imperceptible information within the generated text,
thereby allowing for efficient detection and tracking of the model's potential abuse.

Watermarking techniques can serve multiple purposes, such as embedding ownership information
within the generated text to protect the intellectual property rights of the model. It can also help
mitigate potential harm caused by LLMs by monitoring where the model is being used and whether it
is being misused or abused.

A good watermarking method should not adversely affect the normal usage of the language model or
degrade the quality of the generated text. However, a prevailing belief holds that there is an inevitable
trade-off between the strength of the watermark and the quality of the output text. For instance,
recent work by Kirchenbauer et al. [32] introduced a method that augmented the logits of a randomly
selected set of "green" tokens. By tuning the "magnitude of logits adjustment", they demonstrated a
trade-off between watermark strength and text quality.

Our primary contribution is to challenge this conventional wisdom. We show that with the right
implementation, watermarking can be accomplished without affecting the output quality. We refer to
this particular type of watermark as an **unbiased watermark**. We approach the problem of output
quality degradation from the perspective of watermark detection. We posit that if the watermark

causes a decline in output quality, there should be a method to guess the presence of the watermark based on the quality. Conversely, if the watermark cannot be detected, it implies that the output quality remains unaffected. Specifically, we provide a proof that with a suitable implementation, watermarking does not affect the output probability distribution. This has significant implications, as users who do not have the private key are unable to discern whether a service provider has applied watermarking to the model. Furthermore, the addition of watermarking does not affect the performance of the generated text in any downstream tasks. **Our main contributions can be summarized as follows:**

- We introduce *unbiased watermark*, an innovative family of watermark methods that guarantee the non-degradation of text quality. In addition, we offer a comprehensive framework that facilitates the design and detection of unbiased watermarks.

- We propose two innovative and practical watermarking techniques known as $\delta$-reweight and $\gamma$-reweight. Through extensive experimentation, we demonstrate that these techniques preserve output quality in machine translation and text summarization tasks.

- We develop an advanced maximin variant of the original log-likelihood ratio test for watermark detection. This novel detection method comes with theoretical guarantees, specifically an upper bound on type I error, thus enhancing the reliability of watermark detection in language models.

## 2  Preliminary

In this section, we delve into the problem of watermarking in the context of LLMs. We begin by setting up the problem and defining essential concepts.

**Problem Modeling:** We first introduce several notations to formalize the problem. Let $\Sigma$ denote the vocabulary set, which is the set of all possible tokens an LLM can generate in a single step. We then define the set $\Sigma^*$ as the collection of all possible strings of any length, including those of length zero.

An LLM generates a sequence of tokens conditioned on a given context. In a single step, the probability of generating the next token $x_{n+1} \in \Sigma$ given the current context, $x_1, x_2, ..., x_n$, can be denoted as $P_M(x_{n+1} \mid x_1, x_2, ..., x_n)$. The LLM operates in an autoregressive fashion, which means the joint probability of generating multiple tokens $x_{n+1}, \ldots, x_{n+m}$ can be written as:

$$P_M(x_{n+1}, \ldots, x_{n+m} \mid x_1, x_2, ..., x_n) = \prod_{i=1}^{m} P_M(x_{n+i} \mid x_1, x_2, ..., x_n, x_{n+1}, \ldots, x_{n+i-1}).$$

For simplicity, we use the following notation: $P_M(\boldsymbol{x}_{n+1:n+m} \mid \boldsymbol{x}_{1:n})$, where $\boldsymbol{x}_{n+1:n+m} = (x_{n+1}, \ldots, x_{n+m}) \in \Sigma^*$.

In the context of watermarking, we introduce a service provider that holds a private key $k$ from the key space $K$. The key $k \in K$ is chosen at random from the prior distribution $P_K(k)$. The watermarked output of the LLM follows distribution $P_{M,w}(x_{n+1} \mid x_1, x_2, ..., x_n; k)$, which is conditioned on both the key $k$ and the context $\boldsymbol{x}_{1:n}$. Similarly, we use the notation $P_{M,w}(\boldsymbol{x}_{n+1:n+m} \mid \boldsymbol{x}_{1:n}; k)$ for the probability of generating a sequence of tokens in a watermarked model.

**Objective.** Our goal is to devise a watermarking scheme that: a) is efficiently detectable by the service provider; b) can't be detected by users and does not negatively impact the quality of the output.

The reason we focus on the detection of watermarks by users is that it is closely related to the output quality. If the watermark causes a degradation in the output quality, there should exist a method to infer the presence of the watermark by examining the quality. Conversely, if the watermark is undetectable, it implies that it does not impact the output quality.

From a statistical testing perspective, a watermark is considered strictly undetectable if the probability distributions of the watermarked and non-watermarked outputs are identical. To capture this notion, we define several desirable properties of watermarking schemes.

**Definition 1** ($n$-shot-undetectable). *For a fixed input sequence $\boldsymbol{a} \in \Sigma^*$, we say that watermarked LLM and key prior pair $(P_{M,w}, P_K)$ is n-shot-undetectable compared to original LLM $P_M$ if*

$$\prod_{i=1}^{n} P_M(\boldsymbol{x}^i \mid \boldsymbol{a}) = \sum_{k \in K} P_K(k) \prod_{i=1}^{n} P_{M,w}(\boldsymbol{x}^i \mid \boldsymbol{a}; k), \quad \text{for any n number of strings } \boldsymbol{x}^i \in \Sigma^*.$$

**Definition 2** (downstream-invariant). *We say the watermarked LLM and key prior pair $(P_{M,w}, P_K)$ are invariant compared to original LLM $P_M$ on downstream tasks iff*

$$\mathbb{E}_{\boldsymbol{x} \sim P_{M,w}(\cdot|\boldsymbol{a};k), k \sim P_K}[f(\boldsymbol{x})] = \mathbb{E}_{\boldsymbol{x} \sim P_M(\cdot|\boldsymbol{a})}[f(\boldsymbol{x})],$$

*for any strings $\boldsymbol{x}, \boldsymbol{a} \in \Sigma^*$, and for any metric $f : \Sigma^* \to \mathbb{R}$.*

Note that the one-shot-undetectable property implies the downstream invariant property. Interestingly, this implication does not require the $n$-shot-undetectable property for $n > 1$, which means a watermarking scheme that is one-shot-undetectable can still maintain the output quality for downstream tasks even if the user might discern the existence of the watermark through multiple generation requests.

In summary, we have outlined the preliminary concepts and objectives for developing a watermarking scheme for LLMs. We highlight the desired properties of $n$-shot-undetectability and downstream invariance, as they provide a rigorous theoretical guarantee of quality preservation and integrity in the deployment of watermark schema. In Section 4, we will present a watermark framework that is provably $n$-shot-undetectable for any given integer $n \geq 1$.

# 3 Warm up: undetectability in a simplified toy environment

In this subsection, we aim to prove the feasibility of undetectability in a highly simplified toy environment. This preliminary analysis serves as a foundation for understanding the more complex scenarios that follow.

**Settings.** Consider a service provider that offers a random number generation service. The service outputs a uniformly distributed random number in the set $\{0, 1\}$. The clean generation process can be represented as $P_M(x) = 1/2, \forall x \in \{0, 1\}$. We assume that the key $k$ belongs to the set $\{0, 1\}$ and is selected with equal probability. With the watermark added, the probability of the new output can be expressed as: $P_{M,w}(x \mid k) = \delta_k(x)$.

Recall that the one-shot-undetectable property can be represented as $P_M(x) = \sum_{k \in K} P_{M,w}(x \mid k) P_K(k)$. Suppose that a user can only make a single request to the service. If the user is unaware of the key, the user will be unable to distinguish whether the received result is watermarked or not. Therefore, in this simplified scenario, the undetectability of the watermark is achieved.

However, there is a considerable gap between this toy example and the practical implementation of watermarking in LLMs. Firstly, the symbol set $\Sigma$ in LLMs is far more complex than the binary set $\{0, 1\}$, and the probability distribution is not uniform. Besides, the generation process in LLMs is autoregressive, which means that more than one symbol are generated iteratively. Furthermore, the toy example does not satisfy the $n$-shot-undetectable property for $n > 1$.

Despite these differences, this simple example provides essential insights that help in understanding the following sections where we address these challenges. The underlying principles of undetectability remain constant, while their application becomes more intricate in a more complex environment.

# 4 Watermarking with unbiased reweighting

In this section, we build upon the intuition from the previous section and extend the approach to LLMs' generation. The section is structured as follows: Section 4.1 introduces a fundamental mathematical tool for addressing the reweighting problem in general discrete probability distributions. Section 4.2 applies the reweighting technique to LLMs. Section 4.3 presents the final framework.

## 4.1 Distribution reweighting

In its most general form, we consider a random watermark code $E$ and a reweight function $R_E : \Delta_\Sigma \to \Delta_\Sigma$, which depends on the random watermark code $E$. The set of all possible probability distributions on the symbol set $\Sigma$ is denoted as $\Delta_\Sigma$, which forms a simplex.

**Definition 3.** *A **reweighting function** is a tuple $(\mathcal{E}, P_E, R)$ where $\mathcal{E}$ is called the watermark code space, $P_E$ is a probability distribution on space $\mathcal{E}$, and $R$ is a function $R : \mathcal{E} \times \Delta_\Sigma \to \Delta_\Sigma$. For a specific watermark code $E \in \mathcal{E}$, we denote the partially evaluated reweighting function as $R_E : \Delta_\Sigma \to \Delta_\Sigma$.*

**Definition 4.** *Given a random watermark code $E$ and a reweighting function $R_E : \Delta_\Sigma \to \Delta_\Sigma$, we say that $R$ is an **unbiased reweighting function** if and only if for all $P \in \Delta_\Sigma$, $\mathbb{E}_E[R_E(P)] = P$.*

### 4.1.1 Existing reweighting methods

Kirchenbauer et al. [32] essentially comprise two reweighting methods in their work, but neither of them satisfies the unbiased property.

Both methods have $\mathcal{E}$ as the set of mappings $f : \Sigma \to \{\text{red}, \text{green}\}$, such that $f$ maps half of the tokens in $\Sigma$ to 'red' and the other half to 'green', and $P_E$ as a uniform distribution. Therefore, the random watermark code $E$ assigns each symbol to either *red* or *green*. The "Hard Red List" method sets the probability of all red symbols to zero and renormalizes the probabilities of the remaining vocabulary. The second method is "Soft Red List" blocking, where they randomly select the same "Red List" as the first method and decrease the corresponding probability for red symbols by adding a constant $\delta$ to the logits of the green symbols, then apply softmax to obtain the final probabilities.

### 4.1.2 Unbiased reweighting methods

In this section, we present two reweighting methods that satisfy the unbiased property.

$\delta$**-reweight:** Let the watermark code space $\mathcal{E}$ be the interval $[0, 1]$, and let $P_E$ be the uniform probability on $\mathcal{E}$. Leveraging *Inverse Transform Sampling*[1] [14], we can sample from distribution $P \in \Delta_\Sigma$ using a uniformly distributed random number in $[0, 1]$. Therefore, we have a mapping $sampling_P : \mathcal{E} \to \Sigma$. The $\delta$-reweight just returns a delta distribution $R_E(P) = \delta_{sampling_P(E)}$.

It is important to note that while the reweighted distribution for each individual random event $E$ is a delta distribution, the mean output token probabilities remain the original distribution $P$ when considering the randomness of $E$.

$\gamma$**-reweight:** Let the watermark code space $\mathcal{E}$ be the set of all bijective function between vocabularies set $\Sigma$ and a set of indices $[|\Sigma|] = \{1, \ldots, |\Sigma|\}$, where $|\Sigma|$ is the size of vocabularies set $\Sigma$. Essentially, any watermark code $E$ is an indexing function for vocabularies set $\Sigma$, and is also equivalent to a total order on $\Sigma$. Let $P_E$ be the uniform probability on $\mathcal{E}$, it is easy to sample a watermark code $E$ by randomly shuffling the symbol list.

Assume the original distribution is $P_T(t) \in \Delta_\Sigma, \forall t \in \Sigma$. Given the watermark code $E : \Sigma \to [|\Sigma|]$, we construct auxiliary functions $F_I(i) = \sum_{t \in \Sigma} \mathbf{1}(E(t) \leq i) P_T(t)$, $F_S(s) = \max(2s - 1, 0)$, $F_{I'}(i) = F_S(F_I(i))$. The $\gamma$-reweight yields new distribution $P_{T'}(t) = F_{I'}(E(t)) - F_{I'}(E(t) - 1)$.

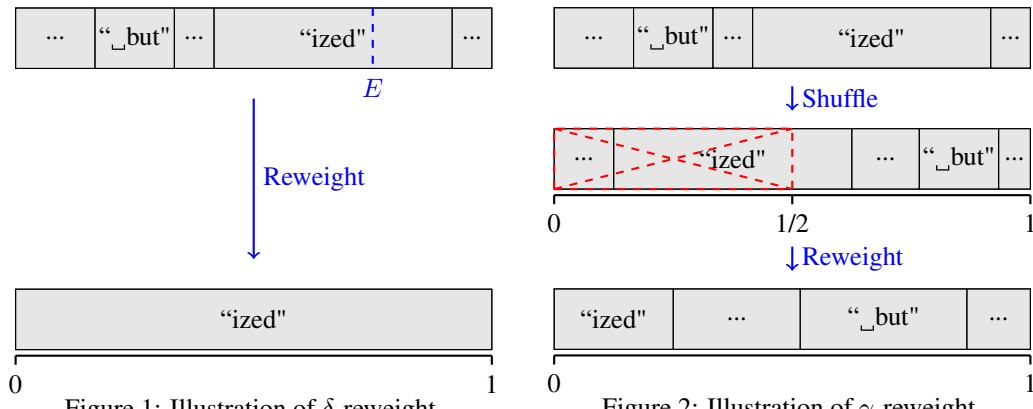

Figure 1: Illustration of $\delta$-reweight.     Figure 2: Illustration of $\gamma$-reweight.

We provide illustrations of the $\delta$-reweight and $\gamma$-reweight methods in Figures 1 and 2. Each block represents a token, and the width represents the probability of that token, so the total length is 1 The left panel shows the $\delta$-reweight method, where each individual random watermark code $E \in [0, 1]$ uniformly sampled from interval $[0, 1]$ corresponds to a specific token according to the horizontal axis, and the reweighted distribution is just a $\delta$ distribution on that token, such that the selected token has 1 probability, and all other vocabulary tokens have a probability of 0. The right panel demonstrates the $\gamma$-reweight method. First, the symbol set is shuffled. Then, the left half of the regions are rejected, and the remaining regions are amplified with a factor of 2.

Both methods are unbiased[1] when considering the randomness of the watermark code $E$. For $\delta$-reweight, we can see that by noticing that the probability of returning a $\delta$ distribution on a token is

---

[1]Detailed definition and rigorous proof can be found in Appendix D

just the original probability on that token, therefore the weighted average of all delta distributions is still the original probability. In the case of $\gamma$-reweight, although certain regions are rejected and the other regions are amplified, every token has the same probability to be in the rejected or amplified region, thus ensuring the unbiased property.

## 4.2 Reweighting for autoregressive model

The reweighting methods presented in the previous section can be applied to single token-generation directly. Given a prefix $\boldsymbol{x}_{1:n}$, the probability distribution for generating a new token without a watermark is denoted as $P_M(\cdot|\boldsymbol{x}_{1:n}) \in \Delta_\Sigma$. For a random watermark code $E$, we sample from a new distribution $P_{M,w}(\cdot|\boldsymbol{x}_{1:n}) = R_E(P_M(\cdot|\boldsymbol{x}_{1:n})) \in \Delta_\Sigma$. If the reweighting function is unbiased, we have $\mathbb{E}_E[R_E(P_M(\cdot|\boldsymbol{x}_{1:n}))] = P_M(\cdot|\boldsymbol{x}_{1:n})$. This ensures that, for an individual unaware of the watermark code, it is impossible to determine whether a new token is sampled directly from $P_M(\cdot|\boldsymbol{x}_{1:n})$ or from $P_{M,w}(\cdot|\boldsymbol{x}_{1:n}; E)$ for a random watermark $E$. However, if the watermark code is known, one can perform statistical hypothesis testing to determine the likelihood of a token being sampled from either distribution.

The main challenge now is constructing the watermark code $E$. Since the LLM generation task is autoregressive, multiple reweighting steps are required, with each step needing a watermark code $E_i$ for reweighting the distribution of token $x_i$.

### 4.2.1 Independence of watermark codes

It is crucial that $E_i$ values are independent to ensure the unbiased nature of the entire sequence, rather than just the single-token generation process.

**Theorem 5.** *Given an unbiased reweighting function $(\mathcal{E}, P_E, R)$, if $E_i$ values are i.i.d. with the distribution $P_E$, we have:* $\mathbb{E}_{E_1,\ldots,E_n}[P_{M,w}(\boldsymbol{x}_{1:n}|\boldsymbol{a}_{1:m})] = P_M(\boldsymbol{x}_{1:n}|\boldsymbol{a}_{1:m})$.

If the $E_i$ values are not independent, we cannot guarantee that the generation probability of the entire sequence remains unbiased. As an extreme example, consider a case where all $E_i$ values are identical. Referring to the random bit example in the previous section, assume that the correct distribution is a sequence where each token is a random 0 or 1 with equal probability. Identical $E_i$ values would result in identical token outputs, ultimately producing sequences consisting solely of 0's or 1's, which is clearly biased.

### 4.2.2 Context code

To construct a large number of independent watermark codes $E_i$ during watermarking and to know the used $E_i$ values during watermark detection, we follow an approach similar to Kirchenbauer et al. [32] by combining the information from the prefix and a secret key to construct $E_i$.

For a single token generation process, given a prefix $x_1, x_2, ..., x_n$, we consider an abstract context code space $C$ and an abstract context code generation function $cc : \Sigma^* \to C$. Based on the prefix, we construct the context code $c_{n+1} = cc(x_1, x_2, ..., x_n)$. Specific examples include using the entire prefix $c_{n+1} = (x_1, x_2, ..., x_n)$, and using the $m$ most recent prefixes $c_{n+1} = (x_{n-m+1}, ..., x_n)$. Our comprehensive framework accommodates diverse context code generation approaches, particularly those that integrate error-correcting mechanisms to augment watermark resilience in the face of text manipulation attacks. Nevertheless, we refrain from delving into these strategies within the confines of this paper and consider it a subject for subsequent investigation.

The final watermark code is defined as $E_i = \hat{E}(c_i, k)$, using a watermark code generation function $\hat{E} : C \times K \to \mathcal{E}$.

**Definition 6.** *Given an unbiased reweighting function $(\mathcal{E}, P_E, R)$ and a context code space $C$, an **unbiased watermark code generation function** is a tuple $(\mathcal{E}, P_E, R, C, K, P_K, \hat{E})$ that satisfies:*

1. *Unbiasedness: $\mathbb{E}_{k \sim P_K}[R_{\hat{E}(c,k)}(P)] = P, \forall P \in \Delta_\Sigma, \forall c \in C$.*

2. *Independence: For any $n$ distinct $c_1, \ldots, c_n \in C$, the values $R_{\hat{E}(c_i,k)}(P)$ are mutually independent.*

**Theorem 7.** *For any unbiased reweighting function and context code space, an unbiased watermark code generation function always exists.*

In practice, pseudorandom numbers can be used to implement the unbiased watermark code generation function in the above theorem. Specifically, the hash value $\mathrm{hash}(c, k)$ can be used as a random seed

to sample $E$ from $P_E$ as an implementation of $E = \hat{E}(c, k)$. In this paper, we employ SHA-256 for hash function and a 1024-bit random bitstring as the key $k$.

An unbiased watermark code generation function ensures that watermark codes $E_i$ are independent with each other if only their context codes are different. During the generation of a sequence, context codes may be repeated, although this is a rare event in practice. If $c_i$ and $c_j$ are equal, then $E_i$ and $E_j$ are also equal, violating the independence of $E_i$. A simple workaround is to skip reweighting for a token when encountering a previously used context code. In other words, we set $P_{M,w}(\cdot|\boldsymbol{a}_{1:m}, \boldsymbol{x}_{1:i-1}) = P_M(\cdot|\boldsymbol{a}_{1:m}, \boldsymbol{x}_{1:i-1})$ if the context code has appeared before.

## 4.3 Framework

---

**Algorithm 1** Watermarking framework

---

1: **Input:** key for watermark $k \in K$, prompt $\boldsymbol{a}_{1:m} \in \Sigma^*$, generate length $n \in \mathbb{N}$, initial code history $cch \in 2^C$, context code function $cc : \Sigma^* \to C$, watermark code generation function $\hat{E} : C \times K \to \mathcal{E}$, and reweighting function $R : \mathcal{E} \times \Delta_\Sigma \to \Delta_\Sigma$.
2: **for** $t = 1, \ldots, n$ **do**
3:     $P_i \leftarrow P_M(\cdot \mid \boldsymbol{a}_{1:m}, \boldsymbol{x}_{1:i-1})$                               ▷ original distribution
4:     $c_i \leftarrow cc(\cdot \mid \boldsymbol{a}_{1:m}, \boldsymbol{x}_{1:i-1})$                               ▷ context code
5:     **if** $c_i \in cch$ **then**
6:         $Q_i \leftarrow P_i$                                       ▷ skip the reweighting
7:     **else**
8:         $cch \leftarrow cch \cup \{c_i\}$                           ▷ record history
9:         $E_i \leftarrow \hat{E}(c_i, k)$                              ▷ watermark code
10:         $Q_i \leftarrow R_{E_i}(P_i)$                           ▷ reweighted distribution
11:     Sample the next token $x_i$ using distribution $Q_i$
12: **return** $\boldsymbol{x}_{1:n}$

---

Integrating the tools discussed earlier, we present a general framework for watermarking here. The algorithm for this framework is outlined in Algorithm 1.

We note that our abstract framework requires the specification of two key components in order to be practically implemented: the unbiased reweight function $R_E$ and the context code function $cc$.

# 5 Statistical hypothesis testing for watermark detection

In the previous section, we discussed the process of adding a watermark to a text based on a secret key $k$ and a given prompt $\boldsymbol{a}_{1:m}$. The watermark-embedded text can be sampled from the distribution $P_{M,w}(\boldsymbol{x}_{1:n}|\boldsymbol{a}_{1:m}; k)$. In this section, we focus on the watermark detection task, which is the inverse problem of watermark embedding.

Given a text $\boldsymbol{x}_{1:n}$, the goal of watermark detection is to infer whether it is more likely to be generated from the unmarked distribution $P_M(\boldsymbol{x}_{1:n}|\boldsymbol{a}_{1:m})$ or the marked distribution $P_{M,w}(\boldsymbol{x}_{1:n}|\boldsymbol{a}_{1:m}; k)$. This problem can be formulated as a statistical hypothesis test between two competing hypotheses: $H_0$, which posits that $\boldsymbol{x}_{1:n}$ follows the unmarked distribution, and $H_1$, which posits that $\boldsymbol{x}_{1:n}$ follows the marked distribution.

## 5.1 Score-based tesing

We focus on a particular kind of score-based testing, which assigns a score to each token in the text. The score can be interpreted as the confidence that the token was generated by the watermark model rather than the original model. Scores $s_i$ can be computed based on $\boldsymbol{x}_{1:i}$, in accordance with the autoregressive manner of the generation process.

The total score $S$ is given by $S = \sum_{i=1}^{n} s_i$. A threshold $\hat{S}$ is set such that if $S < \hat{S}$, the null hypothesis $H_0$ is accepted, indicating insufficient evidence to conclude that the text contains a watermark. Otherwise, the null hypothesis is rejected. There are two types of error probabilities associated with this decision process: Type I error, which is the probability of incorrectly rejecting

the null hypothesis under $H_0$, denoted as $P_{H_0}(S \geq \hat{S})$, and Type II error, which is the probability of incorrectly accepting the null hypothesis under $H_1$, denoted as $P_{H_1}(S < \hat{S})$.

To derive theoretical results, we require the scores to have a specific property: under the null hypothesis $H_0$, the exponential momentum of $s_i$ is bounded, conditioned on the preceding context $\boldsymbol{x}_{1,i-1}$. This requirement leads to an upper bound on $\alpha$, the Type I error probability.

To derive theoretical results, we require that the scores have a particular property: the exponential moment of $s_i$ under $H_0$ should be bounded, conditioned on the previous text $\boldsymbol{x}_{1,i-1}$. This requirement leads to an upper bound on the Type I error rate.

**Theorem 8.** *Given a probability space $(\Omega, \mathcal{A}, P)$ and a $\Sigma$-valued stochastic process $x_i : 1 \leq i \leq n$, as well as an $\mathbb{R}$-valued stochastic process $s_i : 1 \leq i \leq n$, let $\mathcal{F}_i^x := \sigma(x_j \mid 1 \leq j \leq i)$ and $\mathcal{F}_i^s := \sigma(s_j \mid 1 \leq j \leq i)$ be the corresponding filtrations, where $\sigma(\cdot)$ denotes the $\sigma$-algebra generated by random variables. If $\mathcal{F}_i^s \subseteq \mathcal{F}_i^x$ and $\mathbb{E}[\exp(s_i)|\mathcal{F}_{i-1}^x] \leq 1$, then $P(\sum_{i=1}^n s_i \geq t) \leq e^{-t}$.*

Therefore, to ensure that the Type I error probability has an upper bound $\alpha$, we can set the threshold $\hat{S}$ as $\hat{S} = -\log(\alpha)$. In the following, we discuss two special scores.

## 5.2 Log likelihood ratio (LLR) score

According to the Neyman-Pearson lemma, the likelihood ratio test is the most powerful test among all tests with the same Type I error rate. Specifically, the log-likelihood ratio (LLR) score is defined as $s_i = \log \frac{P_{M,w}(x_i|\boldsymbol{a}_{1:m},\boldsymbol{x}_{1:i-1};k)}{P_M(x_i|\boldsymbol{a}_{1:m},\boldsymbol{x}_{1:i-1})}$, and the total score becomes $S = \log \frac{P_{M,w}(\boldsymbol{x}_{1:n}|\boldsymbol{a}_{1:m};k)}{P_M(\boldsymbol{x}_{1:n}|\boldsymbol{a}_{1:m})}$.

We now provide an optimization derivation of the above $s_i$ to gain intuition and set the foundation for the maximin variant of the LLR score in the next section. Let $P_i = P_M(\cdot|\boldsymbol{a}_{1:m}, \boldsymbol{x}_{1:i-1})$, $Q_i = P_{M,w}(\cdot|\boldsymbol{a}_{1:m}, \boldsymbol{x}_{1:i-1}; k)$, and let $s_i = S_i(x_i) \in \mathbb{R}$ denote the score corresponding to different $x_i$. Note that $P_i$, $Q_i$, and $S_i$ are all functions with signature $\Sigma \to \mathbb{R}$, therefore equivalent to vectors of dimension $|\Sigma|$. We can define the inner product as $\langle P_i, S_i \rangle = \sum_{x \in \Sigma} P_i(x) S_i(x)$.

The requirement $\mathbb{E}[\exp(s_i)|\mathcal{F}_{i-1}^x] \leq 1$ can be reformulated as $\langle P_i, \exp(S_i) \rangle \leq 1$, where the exponential function is applied element-wise. Instead of minimizing the Type II error directly, we aim to maximize the average score under $H_1$, i.e., $\langle Q_i, S_i \rangle$.

The optimization problem becomes $\max_{S_i} \langle Q_i, S_i \rangle$, s.t. $\langle P_i, \exp(S_i) \rangle \leq 1$. The optimal solution is given by $S_i(x) = \log \frac{Q_i(x)}{P_i(x)}$, which recovers the optimal log likelihood ratio score.

## 5.3 Maximin variant of LLR score

One major limitation of the LLR score described in the previous section is that when $Q_i(x) = 0$, $S_i(x) = -\infty$. This means that as long as a single token does not come from the watermark model $P_{M,w}$, the score becomes negative infinity, making it impossible to reject the null hypothesis $H_0$.

A more general reason for this issue is that the watermark model $P_{M,w}$ used in the detection process may not exactly match the true distribution of the watermarked text. In practice, potential sources of discrepancy include editing (e.g., a text sampled from $P_{M,w}$ may undergo some degree of editing before being watermark detection) and imperfect estimation of the generation process (e.g., due to lack of knowledge of the exact prompt and temperature used during generation).

To address this problem, we consider a perturbed generation distribution. Instead of the original hypothesis $H_1$, where $\boldsymbol{x}_{1:n}$ follows the watermark distribution $P_{M,w}$, we now assume that $\boldsymbol{x}_{1:n}$ follows a distribution $P'_{M,w}$, which is similar to but not identical to $P_{M,w}$. Specifically, during the generation of each token, the total variation (TV) distance between $Q'_i$ and $Q_i$ is bounded by $d$.

The corresponding new optimization problem is

$$\max_{S_i} \min_{Q'_i \in \Delta_\Sigma, TV(Q'_i, Q_i) \leq d} \langle Q'_i, S_i \rangle, \quad s.t. \langle P_i, \exp(S_i) \rangle \leq 1.$$

Intuitively, the optimal solution for $Q'_i$ in the inner optimization decreases $Q'_i(x)$ when $S_i(x)$ is large and increases $Q'_i(x)$ when $S_i(x)$ is small.

The computation of the maximin solution can be done efficiently in $\widetilde{O}(|\Sigma|)$ time and the specific algorithm is shown in Appendix C.

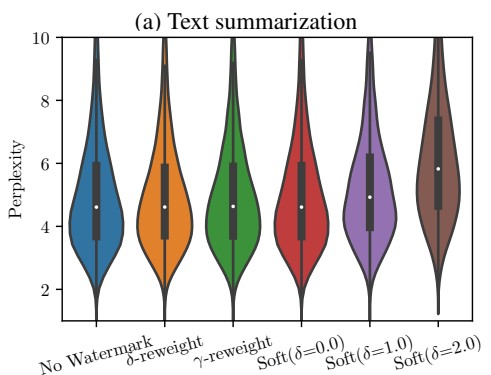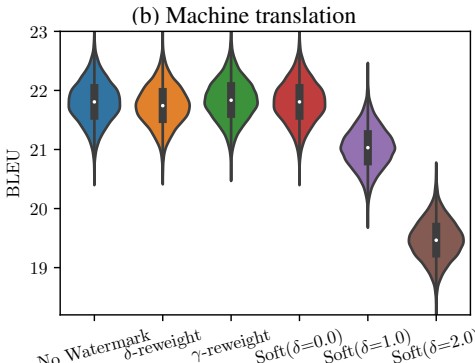

Figure 3: Distribution of perplexity of output for TS and BLEU score for MT.

It is important to note that the maximin variant of the LLR score is more robust than the standard LLR score, as it yields higher scores when the text has undergone some degree of editing. However, it is not specifically designed to defend against any attacks.

A hyperparameter $d \in [0, 1]$ that represent the perturbation strength is introduced in the score. Intuitively, if the text to be detected has undergone more editing and deviates further from the distribution $P_{M,w}$, $d$ should be larger. In practice, we recommend using grid search to select the best value of $d$. Assuming there are $A$ candidate values for $d$, corresponding to $A$ different scores $s_i^{(a)}$ ($1 \leq a \leq A$), we can modify Theorem 8 as follows.

**Theorem 9.** *Under the same conditions as Theorem 8, but with multiple scores $s_i^{(a)}$, we have*

$$P\left( \max_{1 \leq a \leq A} \left( \sum_{i=1}^{n} s_i^{(a)} \right) \geq t \right) \leq A e^{-t}.$$

Thus, when using grid search, the final threshold should be adjusted as $\hat{S} = -\log(\alpha) + \log(A)$. This ensures that the upper bound of the type I error is still $\alpha$.

# 6 Experiments

We evaluate the performance of our Unbiased Watermarks on two important applications of seq2seq models: text summarization (TS) and machine translation (MT). For the TS task, we use the BART-large model [37] and the CNN-DM [25] corpus as our training dataset. The MT task involves translating English to Romanian, for which we employ the Multilingual BART (MBart) [37] model on the WMT'14 En-Ro corpus. For further details on the experiment setup, please refer to Appendix E.

Table 1: Performance of different watermarking methods on TS and MT. We use F1 scores of BERTScore and scale BERTScore and ROUGE-1 with a factor of 100.

|  | Text summarization | | | Machine translation | |
| --- | --- | --- | --- | --- | --- |
|  | BERTScore ↑ | ROUGE-1 ↑ | Perplexity ↓ | BERTScore ↑ | BLEU ↑ |
| No Watermark | $32.70 \pm 0.08$ | $38.56 \pm 0.09$ | $5.024 \pm 0.018$ | $55.9 \pm 0.3$ | $21.8 \pm 0.3$ |
| $\delta$-reweight | $32.71 \pm 0.08$ | $38.57 \pm 0.09$ | $5.022 \pm 0.018$ | $56.3 \pm 0.3$ | $21.7 \pm 0.3$ |
| $\gamma$-reweight | $32.69 \pm 0.08$ | $38.60 \pm 0.09$ | $5.019 \pm 0.018$ | $56.2 \pm 0.3$ | $21.8 \pm 0.3$ |
| Soft($\delta$=0.0) | $32.70 \pm 0.08$ | $38.56 \pm 0.09$ | $5.024 \pm 0.018$ | $55.9 \pm 0.3$ | $21.8 \pm 0.3$ |
| Soft($\delta$=1.0) | $32.35 \pm 0.08$ | $38.20 \pm 0.09$ | $5.313 \pm 0.018$ | $55.1 \pm 0.3$ | $21.0 \pm 0.3$ |
| Soft($\delta$=2.0) | $31.21 \pm 0.08$ | $37.17 \pm 0.08$ | $6.253 \pm 0.022$ | $53.8 \pm 0.3$ | $19.5 \pm 0.3$ |

Our primary focus is to compare the performance of our proposed unbiased watermarking methods including the $\delta$-reweight and $\gamma$-reweight, with the soft-red-list method presented by Kirchenbauer et al. [32]. The strength of the watermark in the soft-red-list approach is controlled by a parameter $\delta$.

The quality of output post-watermarking is presented in Table 1. We observed that the output quality remains unaffected by our unbiased watermark methods, both for the $\delta$-reweight and $\gamma$-reweight,

Table 3: Text sampled from OPT-6.7B, with and without watermarks. For "No watermark" (NW), the score is computed based on $\delta$-reweight. When watermarks are included, the corresponding reweighting function is used for computing score. The optimal perturbation strengths $d$ obtained by grid search are $0.9, 0.0, 0.0$ for three outputs respectively.

| Prompt | What is a watermark? What's the purpose of it? | score | p-value[2] |
|---|---|---|---|
| NW | Why don't you want it on there? I'm confused..\nI think he meant to say logo, since he wrote "watermark", so the first word must be a typo.\nYes! Exactly typo. Sorry, English is not my native language.. Thanks for the explanation! | 0.30 | 8.14 |
| $\delta$-RW | \nIt is supposed to be watermarking the pictures that you took with your phone i think. So, so you can share your pictures and not take credit for them. | 75.9 | 1.2e−32 |
| $\gamma$-RW | \nA watermark is a small image or logo (often in square pixels) that is placed over the larger, original image. It serves primarily to distinguish copyright or ownership of large images (such as banners and logos) and, on rare occasion, to identify small images (such as thumbnail images for blog posts and pictures). | 32.9 | 5.7e−14 |

irrespective of the task and metric. Conversely, the soft-red-list method, when $\delta = 0$, does not introduce any watermark and hence does not affect output quality. However, for $\delta > 0$, it significantly deteriorate the quality of output.

Figure 3 provides a more intuitive depiction of the score distributions. It is evident that our unbiased watermark methods not only ensure that the mean performance remains unaffected but also that the performance distribution is stable. Conversely, the soft-red-list method shows a notable performance decrease.

In terms of watermark detection, we compute score associated with each token. The mean and variance of score per token for TS and MT are presented in Table 2. As a heuristic, if the sum of the scores for all tokens in a sentence reaches $10$, a p-value of less than $0.0005$ is ensured. If the sum score hits $20$, the p-value must be less than $3\mathrm{e}{-8}$.

Table 2: Mean and variance of score per token for different reweighting methods and different tasks.

| | Text summarization | Machine translation |
|---|---|---|
| $\delta$-RW | $0.8784 \pm 1.4354$ | $0.4192 \pm 1.1361$ |
| $\gamma$-RW | $0.2207 \pm 0.3678$ | $0.1056 \pm 0.2916$ |

Additionally, we provide an example of watermarking applied to a completion task in Table 3. It visually demonstrates the score distribution across tokens: positive scores are represented in green, and negative ones in red. The intensity of the color corresponds to the magnitude of the score, with darker shades representing larger absolute values.

# 7 Related work

The idea of watermarking text has been widely explored by many researchers [11, 31, 44, 45, 4, 28, 49, 43], even before the advent of large language models. Several techniques involve editing existing text to add a watermark, such as changing synonyms [54, 57, 9, 59, 66] or visually indistinguishable words [46], altering sentence structures [56, 55, 38], and employing neural networks [22, 23, 67].

Recent advancements in generative models have opened new possibilities for directly generating watermarked results. Two relevant works in this domain are by Kirchenbauer et al. [32] and Aaronson [1]. Due to space constraints, we moved the in-depth analysis and other related work to Section B.

# 8 Conclusion

Overall, this paper provides a novel framework of watermarking for language models, demonstrating that it is possible to use watermark to protect intellectual property and monitor potential misuse without compromising the quality of the generated text. This research serves as a valuable foundation for future work in the field of watermarking for large language models.

---

[2]This is an upper bound computed based on Theorem 9. The upper bound could be larger than 1, but this does not necessarily imply that the p-value exceeds 1.

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
