# OpenReview forum: "Unbiased Watermark for Large Language Models"
_NeurIPS.cc/2023/Conference — Submitted to NeurIPS 2023_

### Official Review · Reviewer_WEht · 2023-07-04

**Soundness:** 2 fair
**Presentation:** 2 fair
**Contribution:** 2 fair
**Rating:** 3
**Confidence:** 4

**Summary:**

This study explores watermarking large language models without reducing output quality. It introduces "unbiased watermarking" which avoids trade-offs in prior work. Two novel techniques - $\delta$-reweight and $\gamma$-reweight - are proposed along with an improved likelihood ratio test for detection. Risks of large language models and how unbiased watermarking enables responsible AI are discussed.

**Strengths:**

1. Introduces unbiased watermarking that maintains output quality. Prior work showed a trade-off, but an unbiased watermark avoids this.

2. Proposes novel $\delta$-reweight and $\gamma$-reweight techniques that preserve output quality in experiments.

3. Develops an improved detection method with a proven upper error bound, improving detection reliability.

4. Concrete demonstration of the efficacy of watermarking techniques in maintaining the utility of LLMs for downstream tasks.


**Weaknesses:**

1. The paper lacks a thorough examination of the efficacy of the watermark detection method. It would strengthen the findings to provide more details on factors like detection accuracy, robustness to interference, and computational efficiency of the detection approach.

2. The experiments only test the watermarking techniques on BART language models, without evaluating other popular LLMs like GPT models.

3. Conducting additional experiments using LLMs for other natural language tasks, beyond text summarization and machine translation studied in the paper, would provide a wider test case set and bolster the claims regarding the output quality preservation of the watermarking techniques.

4. The paper does not discuss the resilience of the proposed watermarking methods against potential adversarial attacks or interference attempts.


**Questions:**

What kinds of decoding methods are suitable for this watermarking method?

**Limitations:**

See the weakness section.

---

> ### Author Rebuttal · Authors · 2023-08-10
>
> Thank you for your comprehensive review and valuable feedback on our paper.
>
> We appreciate your recognition of our novel introduction of "unbiased watermarking" and the importance of ensuring output quality. We're glad you've noted the improvements we've offered over previous techniques, as well as our development of an enhanced detection mechanism.
> > The paper lacks a thorough examination of the efficacy of the watermark detection method. It would strengthen the findings to provide more details on factors like detection accuracy
>
> We understand your concerns about the lack of explicit discussion on detection accuracy. While we indirectly provided information on detection accuracy through the reported score per token, we acknowledge that the data can be represented more intuitively.
>
> To address your feedback, we plan to include additional visualizations such as ROC curves and provide computations for the AUC, which will more directly illustrate the detection accuracy. If you have any specific suggestions on other ways we can more effectively demonstrate this information, we would be glad to consider incorporating them into the paper.
> > robustness to interference
> > The paper does not discuss the resilience of the proposed watermarking methods against potential adversarial attacks or interference attempts.
>
> While our main focus in this paper was on introducing and establishing the concept of unbiasedness, we recognize your point about the absence of robustness discussions.
>
> We have now added new experiments to test the robustness of the two unbiased watermarking methods in this paper, and “Soft Red List” method. We truncate the generated text length to 16, and about $\epsilon$ portion of the output is changed to random tokens. For 512 independent samples, we compute the AUC of different watermarking detection methods.
>
> All methods tested here are prone to some level of degradation in performance with increased perturbations. As mentioned in Section G. Limitations, we acknowledge the significance of robustness of watermarking, but we believe that unbiasedness and robustness are two separate research directions. There is existing literature, such as the work by Kirchenbauer et al. 2023 [2], Krishna et al. 2023 [3], Sadasivan et al. [4], which is dedicated to the topic of robustness. A complete response to the watermarking robustness issue requires establishing a threat model, gauging the intensity of attacks, and experimentally verifying watermark hyperparameters. We hope the reviewer could allow us to keep the focus of this paper on introducing and validating the concept of unbiasedness, and leave robustness for other specialized research works.
> > computational efficiency
>
> We can provide further details on the efficiency of our watermarking methods. The computational complexity of the $\delta$-reweight and $\gamma$-reweight is $O(BS∣\Sigma∣)$, where $S$ is the sequence length, $B$ is the batch size, and $∣\Sigma∣$ represents the size of the vocabularies. For the maximin variant of LLR score, the complexity is  $\widetilde{O}(BS∣\Sigma∣)$. When compared to the internal computation of large language models, the computational overhead of the watermark layer is insignificant.
> > 2.The experiments only test the watermarking techniques on BART language models, without evaluating other popular LLMs like GPT models.
>
> We've broadened our experiments to include other popular models like T5 for translation and LLaMA 2 for summarization and poem generation. As GPT-3.5 and GPT-4 are popular, they are proprietary and we don't have access to their full output distributions, thus we are unable to watermark them.
>
> We want to emphasize that the methods proposed in this paper are general and applicable to any auto-regressive language model, not just specific ones such as BART, as our unbiased property is established on the general properties of auto-regressive language models.
>
> > 3.Conducting additional experiments using LLMs for other natural language tasks, beyond text summarization and machine translation studied in the paper, would provide a wider test case set and bolster the claims regarding the output quality preservation of the watermarking techniques.
>
> We've added a new task of generating poetry using LLaMA 2 and evaluated the output using the Perplexity.
>
> If you have specific downstream tasks in mind that you would like to see evaluated, we would welcome your suggestions and are open to incorporating these into our experiments.
> > What kinds of decoding methods are suitable for this watermarking method?
>
> Deciding on a suitable decoding method is still an open question, largely due to the trade-offs between likelihood, diversity, and informativeness [5].
>
> However, focusing on whether the quality of the output is maintained when combining a decoding method with unbiased watermarking method, we find the following:
> - For decoding methods that sequentially modify the probability distribution of each token, e.g. **Top-k Sampling** and **Nucleus Sampling**, we can apply the decoding method first to adjust the probabilities, followed by the watermarking method. This ensures the output quality is equivalent to the use of the decoding method alone.
> - If the decoding method respects the original distribution through sampling (e.g. **Multinomial sampling**, **Arithmetic Sampling**[6]), we can apply the watermarking method first, then use the decoding method. This also maintains the same output quality.
> - However, for deterministic decoding methods like **Beam Search**, we cannot guarantee that combining it with our watermarking method will preserve the output quality. For example, if $\delta$-reweight and Beam Search are used, the Beam Search would make no difference as the $\delta$-reweight has already adjusted the output distribution to $\delta$ distribution.
>
> Once again, we appreciate your thoughtful review and feedback on our paper. Please let us know if you have any additional questions or suggestions.

---

> > ### Author Response · Authors · 2023-08-10
> > **Addition Response from Authors**
> >
> > Robustness experiment:
> >
> > |   $\epsilon$  |      0.0      |      0.1      |      0.2      |      0.3      |      0.4      |      0.5      |
> > |-|-|-|-|-|-|-|
> > |  $\delta$-reweight | 0.9997±0.0005 | 0.9569±0.0021 | 0.8881±0.0043 | 0.8152±0.0059 | 0.7487±0.0056 | 0.6851±0.0067 |
> > |  $\gamma$-reweight | 0.9936±0.0016 | 0.9297±0.0030 | 0.8391±0.0018 | 0.7574±0.0054 | 0.6942±0.0107 | 0.6502±0.0068 |
> > | Soft($\delta$=1.0) | 0.8446±0.0069 | 0.7871±0.0081 | 0.7339±0.0110 | 0.6741±0.0119 | 0.6334±0.0084 | 0.5859±0.0079 |
> > | Soft($\delta$=2.0) | 0.9705±0.0030 | 0.9239±0.0070 | 0.8680±0.0088 | 0.7956±0.0110 | 0.7312±0.0121 | 0.6561±0.0124 |
> >
> > Downstream quality experiment:
> >
> > | |Text summarization (Llama 2), ROUGE-1| Machine translation (T5), BERTScore | Poetry generation (Llama 2), PPL |
> > |--|--|--|--|
> > |$\delta$-reweight| 0.3704±0.0009| 0.577±0.003| 2.71±0.06|
> > |$\gamma$-reweight| 0.3704±0.0009| 0.576±0.003| 2.71±0.08|
> > |Soft($\delta$=0.0)| 0.3705±0.0009| 0.575±0.003| 2.73±0.08|
> > |Soft($\delta$=1.0)| 0.3678±0.0009| 0.571±0.003| 3.04±0.13|
> > |Soft($\delta$=2.0)| 0.3610±0.0009| 0.560±0.003| 3.92±0.16|
> >
> >
> > [1] Kirchenbauer, John, et al. "A watermark for large language models." arXiv preprint arXiv:2301.10226 (2023).
> >
> > [2] Kirchenbauer, John, et al. "On the Reliability of Watermarks for Large Language Models." _arXiv preprint arXiv:2306.04634_(2023).
> >
> > [3] Krishna, Kalpesh, et al. "Paraphrasing evades detectors of ai-generated text, but retrieval is an effective defense." _arXiv preprint arXiv:2303.13408_ (2023).
> >
> > [4] Sadasivan, Vinu Sankar, et al. "Can ai-generated text be reliably detected?." _arXiv preprint arXiv:2303.11156_ (2023).
> >
> > [5] Zarrieß, Sina, Henrik Voigt, and Simeon Schüz. "Decoding methods in neural language generation: a survey." Information 12.9 (2021): 355.
> >
> > [6] Vilnis, Luke, et al. "Arithmetic Sampling: Parallel Diverse Decoding for Large Language Models." _International Conference on Machine Learning_. PMLR, 2023.

---

> > > ### Author Response · Authors · 2023-08-21
> > > **Official Comment by Authors**
> > >
> > > We noticed that your score for our paper decreased following our rebuttal. We are keen to understand the reasons behind this change, especially since we believed we had addressed the concerns you previously raised.
> > >
> > > Could you kindly provide clarification or specific feedback behind this change?
> > >
> > > Thank you for your time and expertise.

---

### Official Review · Reviewer_qjwP · 2023-07-07

**Soundness:** 2 fair
**Presentation:** 2 fair
**Contribution:** 3 good
**Rating:** 5
**Confidence:** 4

**Summary:**

The paper proposes a modification of the watermark of Kirchenbauer et al. that ensures each next token prediction is marginally indistinguishable from a regular sample from the language model (whereas Kirchenbauer et al. bias some tokens over others). The main idea is to use inverse transform sampling to sample the text token (the paper also proposes a "soft" version of inverse transform sampling, i.e., \gamma-reweight), where the inputs to the sampler are a function of the context.

**Strengths:**

The paper proposes a neat, simple modification of the watermark of Kirchenbauer et al. that addresses a salient problem (i.e., the watermark of Kirchenbauer et al. does not preserve the original text distribution). The paper validates the proposed watermark with both theory and experiments.

**Weaknesses:**

The empirical validation of the watermarked proposed in the paper is somewhat lacking. For example, how robust is the watermark to paraphrasing compared to the watermark of Kirchenbauer et al.? Given Kirchenbauer et al. evaluate robustness to various kinds of paraphrasing attacks, it seems reasonable to expect the paper to do the same. It's also not clear what the purpose of Table 1 and Figure 3 is if the main claim is that the watermark *provably* does not bias the text distribution (i.e., shouldn't all the metrics stay the same?). Also, Section 4.1 seems needlessly abstract, since ultimately both watermarks are variations of inverse transform sampling (it feels unintuitive to think \delta-reweighting as a "reweighting" of a distribution, since it is essentially deterministic).

**Questions:**

Line 88: "Note that the one-shot-undetectable property implies the downstream invariant property." It is not immediately clear why this is the case. In general, both Definitions 1 and 2 would benefit from some more context and/or clearer presentation.

**Limitations:**

The paper does not discuss limitations (if any) of the proposed methods in meaningful detail. The Conclusion section would benefit from more detailed/concrete discussion and takeaways.

---

> ### Author Rebuttal · Authors · 2023-08-10
>
> Thank you for taking the time to review our paper and for recognizing the saliency of the problem that our study addresses.
> > The empirical validation of the watermarked proposed in the paper is somewhat lacking.
>
> We've expanded the empirical validation with a new robustness experiment, and in addition to the original models BART and OPT, we've tested on new models including LLaMA2 and T5. We've also incorporated a new task, poetry generation, and a new metric, GPTScore. We hope these additional experiments address your concern.
>
> > The paper should also evaluate various paraphrasing attacks, similar to Kirchenbauer et al.
>
> Indeed, the original paper doesn't contain a discussion on robustness, as the main focus of our paper is on unbiasedness.
>
> However, to address your point, we have now added new experiments to test the robustness of the two unbiased watermarking methods in this paper, and the “Soft Red List” method in Kirchenbauer et al. 2023 [1]. We truncate the generated text length to 16, and about $\epsilon$ portion of the output is changed to random tokens. For 512 independent samples, we report the AUC of different watermarking detection methods:
>
> |   $\epsilon$  |      0.0      |      0.1      |      0.2      |      0.3      |      0.4      |      0.5      |
> |-|-|-|-|-|-|-|
> |  $\delta$-reweight | 0.9997±0.0005 | 0.9569±0.0021 | 0.8881±0.0043 | 0.8152±0.0059 | 0.7487±0.0056 | 0.6851±0.0067 |
> |  $\gamma$-reweight | 0.9936±0.0016 | 0.9297±0.0030 | 0.8391±0.0018 | 0.7574±0.0054 | 0.6942±0.0107 | 0.6502±0.0068 |
> | Soft($\delta$=1.0) | 0.8446±0.0069 | 0.7871±0.0081 | 0.7339±0.0110 | 0.6741±0.0119 | 0.6334±0.0084 | 0.5859±0.0079 |
> | Soft($\delta$=2.0) | 0.9705±0.0030 | 0.9239±0.0070 | 0.8680±0.0088 | 0.7956±0.0110 | 0.7312±0.0121 | 0.6561±0.0124 |
>
> All methods tested here are prone to some level of degradation in performance with increased perturbations. As mentioned in Section G. Limitations, we acknowledge the significance of robustness of watermarking, but we believe that unbiasedness and robustness are two separate research directions. There is existing literature, such as the work by Kirchenbauer et al. 2023 [2], Krishna et al. 2023 [3], Sadasivan et al. [4], which is dedicated to the topic of robustness. A complete response to the watermarking robustness issue requires establishing a threat model, gauging the intensity of attacks, and experimentally searching for the best watermark hyperparameters. We hope the reviewer could allow us to keep the focus of this paper on introducing and validating the concept of unbiasedness, and leave robustness for other specialized research works.
>
> > It's also not clear what the purpose of Table 1 and Figure 3 is if the main claim is that the watermark _provably_ does not bias the text distribution (i.e., shouldn't all the metrics stay the same?).
>
> You're right in pointing out that for all unbiased watermarks, all metrics should remain the same. However, please note that Soft($\delta$=1.0) and Soft($\delta$=2.0) presented in Table 1 and Figure 3 are "Soft Red List" method from Kirchenbauer et al. 2023 [1], which is biased. Hence there are statistically significant differences. On the other hand, there is no statistically significant difference among unbiased watermark methods.
>
> > Also, Section 4.1 seems needlessly abstract, since ultimately both watermarks are variations of inverse transform sampling (it feels unintuitive to think \delta-reweighting as a "reweighting" of a distribution, since it is essentially deterministic).
>
> We appreciate your feedback on this section. We believe that the general framework of the unbiased reweighting function is important to showcase the infinite possibilities of unbiased watermark methods. However, your feedback suggests that our presentation may not have been clear enough. We are open to revising this section, emphasizing the deterministic nature of the $\delta$-reweight, and further clarifying the usage of inverse transform sampling.
>
> > Line 88: "Note that the one-shot-undetectable property implies the downstream invariant property." It is not immediately clear why this is the case. In general, both Definitions 1 and 2 would benefit from some more context and/or clearer presentation.
>
> Thanks for your suggestion. One-shot-undetectable property ensures that distribution is the same, therefore the expectation is also the same in downstream invariant property.  We'll include an explicit proof in the appendix, and, if accepted, use the extra space to provide more context to these definitions, ensuring clarity for the reader.
>
> Once again, we're grateful for your valuable feedback. We hope our responses address your concerns, and we're willing to make the necessary modifications to improve the paper.
>
> [1] Kirchenbauer, John, et al. "A watermark for large language models." arXiv preprint arXiv:2301.10226 (2023).
>
> [2] Kirchenbauer, John, et al. "On the Reliability of Watermarks for Large Language Models." _arXiv preprint arXiv:2306.04634_(2023).
>
> [3] Krishna, Kalpesh, et al. "Paraphrasing evades detectors of ai-generated text, but retrieval is an effective defense." _arXiv preprint arXiv:2303.13408_ (2023).
>
> [4] Sadasivan, Vinu Sankar, et al. "Can ai-generated text be reliably detected?." _arXiv preprint arXiv:2303.11156_ (2023).
>
> Additional Experiment Results:
>
> | |Text summarization (Llama 2), ROUGE-1| Machine translation (T5), BERTScore | Poetry generation (Llama 2), PPL | Machine translation (mbart), GPTScore |
> |--|--|--|--|--|
> |$\delta$-reweight| 0.3704±0.0009| 0.577±0.003| 2.71±0.06|1.25 ± 0.01|
> |$\gamma$-reweight| 0.3704±0.0009| 0.576±0.003| 2.71±0.08|1.26 ± 0.01|
> |Soft($\delta$=0.0)| 0.3705±0.0009| 0.575±0.003| 2.73±0.08|1.26 ± 0.01|
> |Soft($\delta$=1.0)| 0.3678±0.0009| 0.571±0.003| 3.04±0.13|1.31 ± 0.01|
> |Soft($\delta$=2.0)| 0.3610±0.0009| 0.560±0.003| 3.92±0.16|1.41 ± 0.01|

---

> > ### Comment · Reviewer_qjwP · 2023-08-13
> >
> > I appreciate the authors' efforts in crafting their rebuttal. They promise to make various parts of the paper clearer, add more experiments, and also add more theoretical results. While the proposed additions/changes seem reasonable, it is difficult to evaluate whether they will be effective without actually seeing the revised paper. I will not be changing my score (i.e., borderline accept), and would encourage the authors to perhaps consider resubmitting a revised version of the paper to another conference.

---

> > > ### Author Response · Authors · 2023-08-13
> > > **Official Comment by Authors**
> > >
> > > Firstly, we sincerely understand your position in wanting to see the new information in the rebuttal incorporated into a revised paper. We recognize the challenges this poses.
> > >
> > > To emphasize, there aren't new theoretical results being introduced. Our original assertion of preserving the distribution remains sound and unchanged.
> > >
> > > The supplementary experiments and clarifications were introduced based on your feedback to answer your question about robustness and to clear potential misunderstandings. The new experiment is already conducted and results are provided for your consideration. These are minor modifications that do not alter the main conclusion of our study: that we've addressed a salient problem of preserving the output distribution and quality with unbiased watermark algorithm.
> > >
> > > We sincerely thank you for your insights and diligence throughout the review process. We hope that the detailed explanations provided in our previous comments alleviate your concerns, even though these details cannot be incorporated into manuscript during this discussion phase.
> > >
> > > However, should you have any lingering doubts or questions about our methods' effectiveness in addressing the salient problem of preserving the output quality/distribution, please do let us know. We hope our endeavors and contributions resonate with the importance of the problem we solve. Thank you once again for your time and insight.

---

### Official Review · Reviewer_rGHj · 2023-07-07

**Soundness:** 3 good
**Presentation:** 2 fair
**Contribution:** 3 good
**Rating:** 5
**Confidence:** 3

**Summary:**

This paper discuss about the important problem of how to watermark the outputs from language models while keeping the model not impacted by watermarking. A perfect watermark scheme should be undetectable without prior information and should have no harm on the utilities of LLMs. This paper proposes some desired properties of watermark schemes such as **n-shot-undetectable** and **downstream-invariant**. The papers also gives the proof that there exists perfect watermark schemes. And guided by the proposed concepts, two reweighing watermark scheme are proposed, as well as corresponding methods for verification. The experiment results show the proposed methods have minor impact on the generated text quality.


**Strengths:**

1. It's super important to have a theoretical framework to guide researchers to design better watermark schemes, and this paper is one of the pioneers in this direction.
2. Considering those automatic metrics, the results shown in the paper, the proposed two methods look good.

**Weaknesses:**

Major concerns:
1: The quality of the watermarked texts are only evaluated by automatic metrics.
2: The results only comes from BART ( along with examples from OPT ).

Minor concerns:
1: The word 'unbiased' is a little misleading.
2: It's a little hard to understand some paragraphs in this paper.

**Questions:**

1: As I stated before, if some evaluation results from LLMs such as GPTScore. The results would be more convincing.
2:  It's about the 2nd weakness. The generation power of OPT is poor compared to many other LLMs.  Actually, all the provided examples in the paper is not good to me, either for those from a watermarked model or those from a model without watermark. I'm wondering the results from other LLMs such as LLaMA and T5.  Will the quality be severely degraded?
3: The so-called strength of watermark doesn't mean the degradation of generation quality of language models. For examples, texts from a  watermarked model may have a different style from the model without watermark. Both the two styles of texts can be good to us.


**Limitations:**

Please read last section.

---

> ### Author Rebuttal · Authors · 2023-08-10
>
> We're deeply grateful for your acknowledgment of our work's importance and pioneering status. Your recognition is invaluable to us. The following addresses each point in your feedback.
> > 1: The quality of the watermarked texts are only evaluated by automatic metrics.
>
> > 1As I stated before, if some evaluation results from LLMs such as GPTScore. The results would be more convincing.
>
> Indeed, we rely on automatic metrics for evaluating the quality of the watermarked texts. Our theoretical guarantee on the unbiasedness of the text distribution implies that the quality should remain consistent across all metrics, including non-automatic ones. The challenges associated with manual evaluation, such as high costs, subjectivity, and reproducibility, made us prioritize automatic metrics.
>
> On the other hand, we have supplemented our evaluations with the GPTScore metric using text-curie-001 as backend. The result for machine translation are as follows:
> - $\delta$-reweight: 1.25 ± 0.01
> - $\gamma$-reweight: 1.26 ± 0.01
> - No watermark: 1.26 ± 0.01
> - Soft($\delta$=1.0): 1.31 ± 0.01
> - Soft($\delta$=2.0): 1.41 ± 0.01
>
> The GPTScore experiment are consistent with our initial findings, confirming the **downstream-invariant** property.
> > 2: The results only come from BART (along with examples from OPT).
> > I'm wondering the results from other LLMs such as LLaMA and T5. Will the quality be severely degraded?
>
> Our method is designed to work for any auto-regressive language model, not limited to BART and OPT. We opted for BART and OPT in our experiments because they are representative models for encoder-decoder and decoder-only structures.
>
> However, responding to your suggestion, we have now also evaluated T5 for translation tasks and LLaMA for summarization and poem generation. The downstream-invariant property is also observed in these new models: The difference in output quality between no-watermark and biased-watermark versions is statistically significant, but there is no statistically significant difference between the unbiased watermark and the no-watermark versions (, here Soft($\delta$=0.0) contains no watermark).
>
> | |Text summarization (Llama 2), ROUGE-1| Machine translation (T5), BERTScore |
> |--|--|--|
> |$\delta$-reweight| 0.3704±0.0009| 0.577±0.003|
> |$\gamma$-reweight| 0.3704±0.0009| 0.576±0.003|
> |Soft($\delta$=0.0)| 0.3705±0.0009| 0.575±0.003|
> |Soft($\delta$=1.0)| 0.3678±0.0009| 0.571±0.003|
> |Soft($\delta$=2.0)| 0.3610±0.0009| 0.560±0.003|
>
> It's worth emphasizing that our theoretical results are built on the general properties of auto-regressive language models, and not on the specifics of individual models.
> > 1: The word 'unbiased' is a little misleading. 2: It's a little hard to understand some paragraphs in this paper.
>
> Thank you for your suggestion on presentation. We have considered other terms, including "undetectable watermark" and "watermark without performance degeneration." But since the latter two are implications of the “unbiased”, we decided to highlight the property of unbiasedness.
>
> Additionally, we value clarity and are actively working on refining our presentation to ensure better comprehension. If there are specific sections you found hard to understand, please kindly let us know. We would be happy to provide further explanations and clarification.
> > 3: The so-called strength of watermark doesn't mean the degradation of generation quality of language models. For examples, texts from a watermarked model may have a different style from the model without watermark. Both the two styles of texts can be good to us.
>
> We agree there could be different styles of generation between biased-watermark and no-watermark models. However, based on common definitions of what's considered "good," we found that biased watermarking methods led to statistically significant deteriorations on metrics such as PPL, BERTScore, ROUGE score, BLEU score and GPTScore. To our knowledge, we're yet to encounter a metric where a biased watermark leads to a statistically significant improvement. Thus, while we acknowledge style variations, our findings lean towards the quality degradation introduced by biased watermarks.
>
> Again, thank you for your constructive feedback and your valuable suggestions. We hope we have addressed your concerns and made necessary improvements to the paper. Please don't hesitate to reach out if you have additional questions or suggestions.

---

### Official Review · Reviewer_3JdJ · 2023-07-07

**Soundness:** 2 fair
**Presentation:** 3 good
**Contribution:** 3 good
**Rating:** 6
**Confidence:** 2

**Summary:**

This paper introduces a novel framework for embedding watermarks into Large Language Models (LLMs) without compromising their output quality. The proposed watermark is designed to be undetectable by LLM users.

A general framework is put forth for incorporating this unbiased watermark into LLMs. This is achieved using two innovative and practical watermarking techniques: $\delta$-rewrite and $\gamma$-reweight.

Experiments conducted on summarization and machine translation tasks demonstrate that these watermarking techniques do not degrade the LLM's output quality, thereby substantiating the effectiveness and practicality of the proposed framework.

**Strengths:**

This paper tackles a critical issue in the Large Language Models (LLMs) field, which is the misuse of LLMs. The authors propose an unbiased watermark and a novel framework for its implementation. The experimental results clearly prove that this watermarking technique works effectively.

The paper's layout is good and easy to follow. It uses clear formulas and theorems that help explain both the problem and the proposed solution. Overall, this is a solid piece of work that contributes significantly to the field.

**Weaknesses:**

This paper does not provide a comparison with any existing watermark baselines. Such comparisons would be beneficial to demonstrate the relative performance of non-unbiased watermark techniques.

Only two types of downstream tasks have been evaluated in the study, which limits the generalizability of the findings. It would enhance the robustness of the results if a broader range of tasks, perhaps using a comprehensive benchmark, were tested.

Additionally, the evaluation of model output quality relies solely on automatic metrics. However, these metrics alone may not be sufficient to provide a comprehensive assessment of output quality. Including more diverse and possibly human-centric evaluation measures or LLM auto evaluator could strengthen the evaluation process.

**Questions:**

Can you provide information on the computational complexity of the proposed watermarking framework?

---

> ### Author Rebuttal · Authors · 2023-08-10
>
> Thank you for recognizing the novelty and significance of our work. We appreciate the time you took to review our paper and the feedback you provided. Here's our response to address your concerns:
> > This paper does not provide a comparison with any existing watermark baselines.
>
> Actually, we did provide a comparison with the watermarking technique discussed in Kirchenbauer et al. 2023 [1]. To the best of our knowledge, this was the only published and comparable paper on watermarking at the time we submitted our paper. Specifically, Figure 3 illustrates the “Soft Red List” method from Kirchenbauer et al. 2023 [1], which we refer to as the “Soft(...)” method. If you think there are other relevant baselines worthy of comparison, please kindly let us know. We value your expertise and would be happy to consider them.
> > The paper only evaluates two downstream tasks, which may limit the generalizability of the conclusions.
>
> We chose text summarization and translation tasks due to their representativeness in the NLP community, and prior to our work, we haven't seen any studies that assess the impact of watermarking on these tasks.
>
> Moreover, based on your feedback, we have now incorporated an additional task: poetry generation using LLaMA 2, which diverges from text summarization and translation in being an open-ended text generation task without any objective performance metric. Here we report the perplexity:
> - $\delta$-reweight: 2.71 ± 0.06
> - $\gamma$-reweight: 2.71 ± 0.08
> - Soft($\delta$=0.0): 2.73 ± 0.08
> - Soft($\delta$=1.0): 3.04 ± 0.13
> - Soft($\delta$=2.0): 3.92 ± 0.16
>
> We emphasize that our primary findings of the unbiased watermark method have theoretical proofs ensuring their applicability across tasks. The empirical validations on the two (now three) tasks were meant to verify our theoretical findings rather than claim generality solely based on them. On the other hand, the nature of our theoretical proof ensures that its properties should hold true across any task, not just the ones we've tested. Nonetheless, we acknowledge your feedback and welcome any specific downstream task recommendations for added evaluations.
> > The quality of model outputs is evaluated only using automated metrics.
>
> Indeed, we primarily relied on automated metrics for evaluation due to their efficiency, cost-effectiveness, consistency, and reproducibility. While human-centric evaluation measures can provide an additional verification of our theory, they are subjective, expensive, and time-consuming. Given the mathematical guarantees backing our framework, we felt automated metrics were sufficient.
>
> However, we are receptive to your feedback and we have supplemented our evaluation with GPTScore [2], an LLM auto evaluator. Utilizing text-curie-001 for our evaluations, the final results for machine translation are as follows (smaller score is desirable):
> - $\delta$-reweight: 1.25 ± 0.01
> - $\gamma$-reweight: 1.26 ± 0.01
> - No watermark: 1.26 ± 0.01
> - Soft($\delta$=1.0): 1.31 ± 0.01
> - Soft($\delta$=2.0): 1.41 ± 0.01
>
> > Can you provide information on the computational complexity of the proposed watermarking framework?
>
> Certainly! The $\delta$-reweight and $\gamma$-reweight both have an asymptotic complexity of $O(BS∣\Sigma∣)$, where $S$ is the sequence length, $B$ is the batch size, and $∣\Sigma∣$ represents the size of the vocabularies. The maximin variant of the LLR score has an asymptotic complexity of $\widetilde{O}(BS∣\Sigma∣)$. The constants involved are quite small, making the time required to add/detect the watermark negligible compared to the internal operations of the LLM.
>
> We hope this addresses all of your concerns. Thank you once again for your valuable feedback, and we look forward to any further suggestions or queries you may have.
>
> [1] Kirchenbauer, John, et al. "A watermark for large language models." arXiv preprint arXiv:2301.10226 (2023).
>
> [2] Fu, Jinlan, et al. "GPTScore: Evaluate as you desire." arXiv preprint arXiv:2302.04166 (2023).

---

> > ### Comment · Reviewer_3JdJ · 2023-08-21
> >
> > Thanks to the authors for the rebuttal! The addition of the new task and the automatic scorer GPT addresses my concerns. I've adjusted my score to 6 (weak accept).

---

> > > ### Author Response · Authors · 2023-08-21
> > > **Official Comment by Authors**
> > >
> > > We sincerely appreciate your time and effort in evaluating our rebuttal and for your constructive feedback throughout the review process.
> > >
> > > Most importantly, we are grateful for your recognition of our novel contribution. Your remarks about our contribution being "solid" and one that "contributes significantly to the field" truly mean a lot to us.
> > >
> > > Thank you once again for your thoughtful review and for your adjusted score.

---

### Decision · Program_Chairs · 2023-09-21

**Decision:**

Reject

**Comment:**

This paper proposes the notion of an unbiased LLM watermark, which does not modify the probability distribution of generated text.
The reviewers' main concerns were about the limited empirical evaluation of the watermark's utility and robustness.
While I think the bias of a watermarking scheme is important to study, utility and robustness seem to be cornerstone properties of such a scheme that require more in-depth evaluation.